# Costs and cost-effectiveness of the Safer Births Bundle of Care intervention in Tanzania health facilities

Boniphace Marwa[1]*, Corinna Vossius[2], Jan Terje Kvaløy[3,4], Benjamin Kamala[5], Pascal Mdoe[5], Hege Langli Ersdal[2,6], Frida N Ngalesoni[7], Albino Kalolo[8], Domenica Morona[9]

1 Department of Public Health and Social Sciences, Catholic University of Health and Allied Sciences, Mwanza, Tanzania, 2 Department of Simulation-based Learning, Stavanger University Hospital, Norway, 3 Department of Research, Section of Biostatistics, Stavanger University Hospital, Norway, 4 Department of Mathematics and Physics, University of Stavanger, Norway, 5 Department of Research, Haydom Lutheran Hospital, Manyara, Tanzania, 6 Faculty of Health Sciences, University of Stavanger, Norway, 7 School of Public Health and Social Sciences, Muhimbili University of Health and Allied Sciences, Dar es Salaam, Tanzania, 8 Department of Public Health, St. Francis University College of Health and Allied Sciences, Morogoro, Tanzania, 9 Department of Parasitology and Entomology, Catholic University of Allied and Health Sciences, Mwanza, Tanzania

* marwaboniphace3@gmail.com

## Abstract

To reach the Sustainable Development Goals 3.1 to 3.4 for maternal, newborn, and child health, effective and cost-effective interventions are required. The Safer Births Bundle of Care (SBBC) intervention aims to improve the quality of intra- and postpartum care. The SBBC Phase I implementation was conducted in 30 health facilities in Tanzania between 2021 and 2023. Outcome data show a significant decrease in perinatal and maternal mortality. This study presents the costs of the SBBC Phase I, and a cost-effectiveness analysis, where the cost per life saved, per life year gained, and per disability-adjusted life year (DALY) averted, and the benefit–cost ratio (BCR) were estimated.

### Methods

Outcome data were based on the estimated number of newborn and maternal lives saved during the project. Cost data were collected retrospectively at the accounting centre at Haydom Lutheran Hospital.

### Results

The total cost for implementing SBBC Phase I was 4,491,204 USD. The cost per life saved was 4,669 USD, per life year gained 80 USD, and per DALY averted 144 USD, and the BCR was 36.

**Data availability statement:** Data can be requested from Haydom Lutheran Hospital, P.O. Box 9000 Haydom, Manyara, Tanzania, Tel. +255(0)27 253 3194/5, Fax +255(0)27 253 3734, E-mail post@haydom.co.tz.

**Funding:** This study was supported by Laerdal Foundation in the form of a grant awarded to BM (940076722) and Laerdal Foundation in the form of a salary for CV. The specific roles of this author are articulated in the 'author contributions' section. The funders had no role in study design, data collection and analysis, decision to publish, or preparation of the manuscript.

**Competing interests:** The authors have declared that no competing interests exists.

## Conclusion

The analysis of SBBC Phase I showed that it was highly cost-effective by World Health Organization classifications. An economic evaluation of SBBC Phase II will provide information about its sustainability and feasibility for further scaling up.

## 1 Introduction

The World Health Assembly Agenda 2024 called for an acceleration towards the achievement of the Sustainable Development Goals (SDGs) reproductive, maternal, newborn, child, and adolescent health [1,2]. SDG 3.1 to 3.4 targets are key focus areas and require effective and cost-effective investments that bring better healthcare for all [3]. The goals for achieving these SDGs are reducing stillbirths threefold, neonatal mortality fourfold and maternal mortality nine fold [1,4]. Previous research indicates that the most cost-effective way to deliver interventions targeted at reproductive health in low- and middle-income countries lies in the provision of clearly defined packages, where bundles of interventions are implemented at the same time [5,6].

### 1.1 The Safer Births Bundle of Care (SBBC)

The SBBC is a continuous quality improvement intervention aiming at improving maternal and newborn outcomes at all stages of labour and early neonatal life [7]. The design of the SBBC intervention was developed based on the experience of the Helping Babies Breathe programme introduced in 2009 at Haydom Lutheran Hospital (HLH) in Tanzania, which was highly cost-effective [8]. The SBBC intervention employs clinical training coupled with simulation-based training using intermittent low-dose-high-frequency training to ensure competency and skills retention among healthcare providers [9]. In addition, innovative tools are used along with a bundle for monitoring foetal and newborn heart rates, facilitating newborn resuscitation, and for the simulation training of postpartum care for mothers and newborns [7,10]. In a feedback loop, mortality is recorded continuously for each facility and communicated back to the respective healthcare staff. The intervention targets labour management, newborn resuscitation, and management of bleeding after birth as main areas. To ensure scalability and that the program is rooted in the national health organization and local facilities, the initial training was organized as a training cascade, where eight national trainers trained 15 regional trainers who again train two so-called facility champions at each facility. These facility champions organize and supervise the simulation-based LDHF training and adjust it according to the data they receive via the feedback loop. Phase I of its implementation was conducted in Tanzania between 2021 and 2023. Thirty health facilities from five regions (Manyara, Tabora, Geita, Shinyanga and Mwanza) were selected based on the high burden of maternal and perinatal mortality and the high volume of deliveries [10]. All three levels of healthcare services, from primary (health centers) to secondary (district hospitals) and tertiary (regional referral hospitals), in both rural and urban settings, were represented. All facilities provided Comprehensive Emergency Obstetric and Basic Newborn Care

services and had separate labour units and operating theaters. The collection of baseline data started in March 2021, while the implementation was rolled out in a stepped-wedge design between June 2021 and December 2023 [10]. No other relevant changes were made to the health systems or treatment recommendations during the study period.

Due to promising midterm results, an expansion of the programme has been launched and is currently performed as Phase II in a total of 150 facilities in the same regions, with the prospect of a nationwide expansion in Phase III [10,11]

Recently, outcome data of Phase I, based on almost 300,000 recorded mother–baby pairs, were published, reporting a significant decrease in perinatal and maternal mortality [10]. However, to inform decision-makers and potential funders, and to fully evaluate the SBBC Phase I intervention, its costs and cost-effectiveness have to be analyzed. This study thus presents the costs stratified into the start-up, running and research costs of the SBBC Phase I and cost-effectiveness analysis, where the cost per life saved, cost per life years gained, and cost per disability-adjusted life year (DALY) averted and the benefit–cost ratio (BCR) is determined.

## 2 Methods

### 2.1 Cost data collection

Cost data were based on secondary data collected retrospectively in March 2024. Data sources were the itemized budget of the project activities and the actual spending on the itemized activities as registered in the financial records. In addition, interviews were conducted with the principal investigator and the head of the HLH accountancy centre. Based on this information, costs were categorized into start-up, running and research costs. Start-up costs comprised costs incurred at the pre-implementation and preparatory phases of the SBBC Phase I intervention, such as initial training or information for stakeholders [12]. Running costs included among others refresher training, continuous low-dose-high-frequency simulation-based training and supervision. Research costs comprised costs directly related to the scientific evaluation of the SBBC intervention, such as data collection or scientific conferences.

Costs were calculated from a programme's perspective. All costs were estimated in Tanzanian Shillings (TZS) and converted to 2020 USD using a mean exchange rate of TZS 2,305/USD provided by the Central Bank of Tanzania [13].

### 2.2 Effectiveness data collection

The number of lives saved was based on prospective data collection by two trained data collectors located at each facility throughout the study. For babies, the variables collected included, among others, fresh stillbirths, newborn deaths within 24 hours and seven days, foetal heart rate monitoring, foetal presentation, birth weight, sex and multiple pregnancies. From the mothers, data were collected on maternal death, age, parity and whether she was referred from another hospital.

### 2.3 Statistics

The number of perinatal and maternal lives saved was calculated based on the findings reported by Kamala and Ersdal et al., [8]. The estimated number of lives saved during the project period was calculated by taking the estimated risk differences between the baseline period and the implementation period and multiplying them by the number of births in the implementation period. The estimated risk differences were calculated as the differences in death incidences between the baseline period and implementation period adjusted for clinically relevant demographic and clinical factors as reported in Kamala and Ersdal et al., [8]. The number of perinatal life years gained was based on the life expectancy at birth in Tanzania as reported by the Tanzanian National Bureau of Statistics, 2022 [14]. The number of maternal life years gained was based on the remaining life expectancy for mothers reported by World Life Expectancy and the National Bureau of Statistics [14,15]. DALYs averted were calculated based on Fox-Rushby and Sassi [16,17], using the following parameters: Discounting rate (r) = 0.03 per year and age weighting with K = 1 and β = 0.04. Disability weights were not applied as we lacked data on morbidity.

## 2.4  Cost-effectiveness analysis

The cost-effectiveness ratio was calculated by dividing the costs by the number of lives saved, life years gained and DALYs averted when comparing the pre- and post-implementation data of the SBBC Phase I intervention. International Dollars were equated to USD. The BCR was calculated based on the value of statistical life years gained of 2,983 USD as stated by the World Health Organization (WHO) [18].

## 2.5  Sensitivity analysis

To assess the robustness of this cost-effectiveness analysis, a one-way sensitivity analysis (SA) was conducted by altering the following parameters (Table 3):

1. We assumed that the number of lives saved was the lower limit of the 95% confidence interval (CI) of perinatal and maternal lives saved, respectively the higher limit.

2. We assumed that labour costs were double, respectively trebled.

3. We assumed that there were no research costs.

## 2.6  Ethical considerations

The SBBC Phase I project has received formal approval from the National Institute of Medical Research (NIMR) for Tanzania with Ref. NIMR/HQ/R.8c/Vol. I/2060 dated 14 June 2022 and amendment Ref. NIMR/HQ/R.8b/Vol. I/1040 dated 8 August 2022 [19]. The extension was made to Ref. NIMR/HQ/R.8a/ Vol.IX/3458, dated 26th February 2025. The CUHAS Directorate of Research and Innovations also authorized the publication of a Manuscript with Research Clearance Number. CREC/832/2024 of 20th March 2025. Local permission to use the costs and outcome data was sought and obtained from HLH management. No consent was sought from the study population as this study used outcome data as secondary data. However, data confidentiality was ensured.

# 3  Results

## 3.1  Lives saved, life years gained and DALYs averted

A total of 297,755 mother–baby pairs were enrolled during SBBC Phase I implementation, and of these, 281,165 mothers and 277,734 babies were included in the final analysis of cost-effectiveness evaluation.

The total estimated number of lives saved during SBBC Phase I was 580 (95% CI 225, 935) perinatal lives, 382 (95% CI 138, 626) maternal lives, and 962 combined lives saved. Based on the life expectancy at birth in Tanzania in 2022 [14] of 65·5 years the number of potential life years gained for newborns was 37,914 years. The calculation of maternal life years gained was based on the average age of reproduction in this study of 25·8 years and the remaining life expectancy for women in Tanzania at this age of 48·1 years, resulting in 18,374 life years gained [15]. Thus, the total number of life years gained was 56,288. The total number of DALYs averted was 31,221 when age weighting and discounting were applied.

## 3.2  Costs of the SBBC intervention

All the costs incurred were borne by the SBBC Phase I project, which was fully funded by the World Bank through the Global Facility Financing. Funds were administered by UNICEF, whereby nine percent of the total funds were charged as overhead. The total cost budgeted for the implementation of SBBC Phase I was 4,491,204 USD, while the actual

spending was 4,489,848 USD, showing an average of 0·03% deviation between budget and expenditure. However, within the cost items, higher deviations occurred. For example, the cost of national and international travel of faculty was reduced by 9·8% due to the pandemic. Funds not spent on some items such as travel expenses were transferred to other items that needed more funds than the amount budgeted.

Table 1 presents the costs drivers according to resource use. Salaries and staff benefits amounted to 35·2% of all project cost, followed by per diems and allowances for training which summed up 18%. Thus, 53·2% of all expenditure was allocated to labour costs. These were followed by office and equipment maintenance (16.6%), travel costs (9.4) and office administration (9.2%). Other costs included scientific documentation and research publications (4.7%), conference/meeting expenses (4.1%) and so forth in reduced proportions Figs 1 and 2.

### 3.3 Costs per life saved, life years gained, DALYs averted and BCR

Table 2 presents the costs per life saved, life years gained and DALYs averted. Costs per life saved were 4,669 USD, and costs per life year gained were 82·5 USD for combined perinatal and maternal outcomes. Costs per DALY averted (combined) were 31·2 USD. To rank the costs of an intervention per DALY, the WHO discerns three categories of cost-effectiveness: highly cost-effective (less than gross domestic product (GDP) per capita); cost-effective (between one and three times GDP per capita); and not cost-effective (more than three times GDP per capita) [18]. As the GDP per capita in Tanzania was 1,193 USD in 2022, the SBBC programme can be ranked as highly cost-effective [19]. Based on the value of a statistical life year of 2,983 USD, the BCR of Phase I was 36.

### 3.4 Sensitivity analysis

A sensitivity analysis (SA) of the number of lives saved was performed by Kamala and Ersdal et al. to assess the impact of modelling choices and assumptions regarding missing outcome data [8]. The results of the one-way SA by altering the outcome and cost parameters as outlined under methods are presented in Table 3. In a worst-case scenario with treble salary costs and a lower limit of lives saved, the programme costs would result in 810 USD per DALY averted and thus still be ranked as highly cost-effective, while costs in the best-case scenario of a higher limit of lives saved and no research costs would amount to 72 USD per DALY.

**Table 1. Costs for Phase I of the SBBC program, stratified according to costs item.**

|  | Costs in USD | Percentage of total costs |
|---|---|---|
| Salary | 1,594,150 | 35·5 |
| Allowances and per-diems during trainings and meetings | 814,551 | 18·1 |
| Travel costs | 423,968 | 9·4 |
| Meetings (foods, venue), office space | 185,523 | 4·1 |
| Utility, office equipment and equipment maintenance | 744,110 | 16·6 |
| Stationaries | 64,542 | 1·4 |
| Renovations/construction of sites | 39,046 | 0·9 |
| Research publications, Article processing charges, scientific conferences, capacity building courses etc. | 212,560 | 4·7 |
| Overheads/administration | 412,755 | 9·2 |
| **Total** | **4,491,204** | **100.0** |

SBBC=Safer Births Bundle of Care; USD=US Dollar

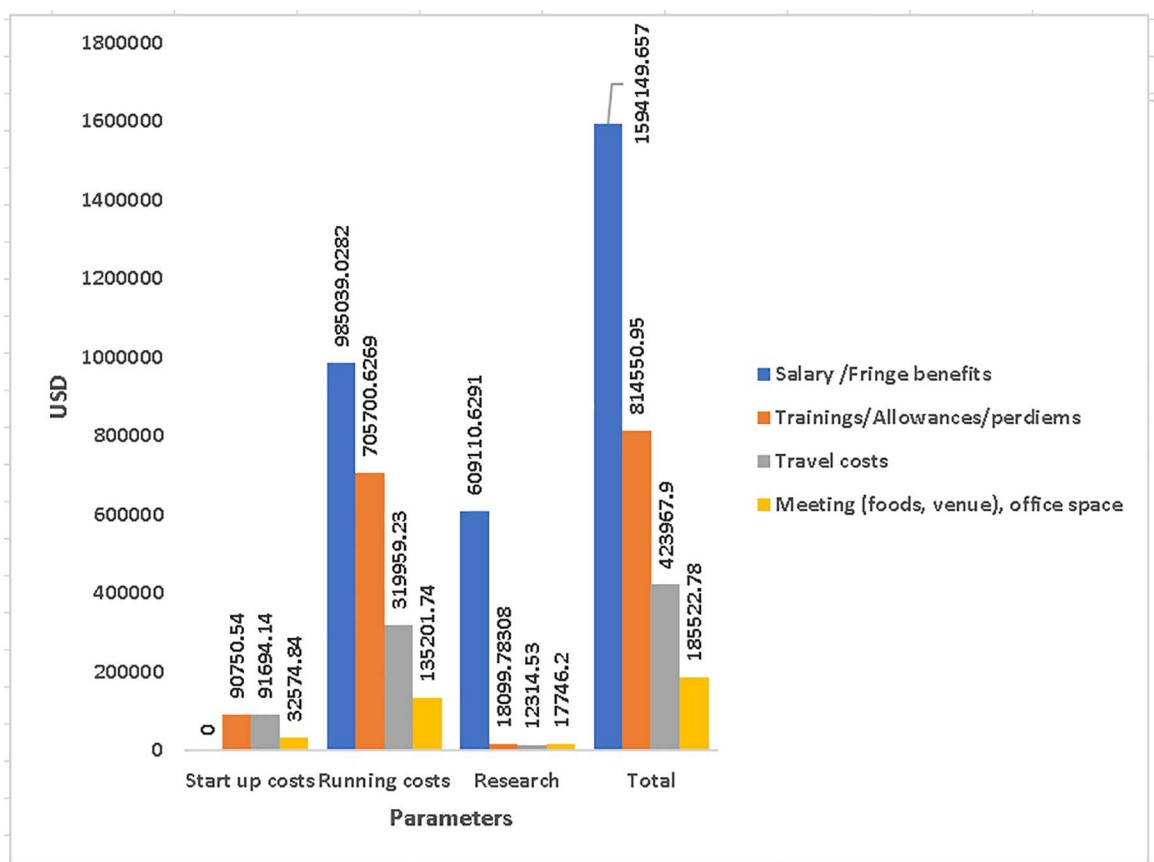

**Fig 1. The four main cost drivers in all cost categories of start-up, running and research costs in the order of higher to lower resource use.**

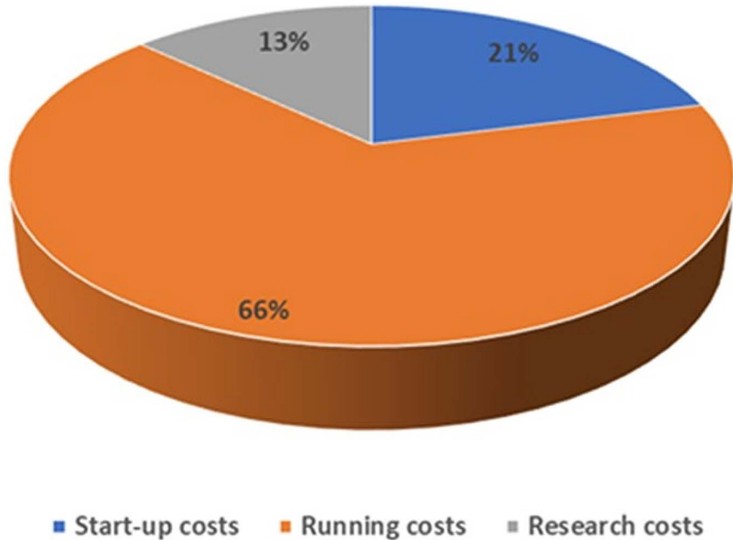

**Fig 2. Costs stratified into start -up cost, running cost, and research cost.**

**Table 2. Cost-effectiveness parameters of the SBBC phase 1 program.**

|  | Number | Costs in USD |
|---|---|---|
| Costs per perinatal life saved | 580 | 7,743 |
| Cost per maternal life saved | 382 | 11,757 |
| Cost per life saved, perinatal and maternal combined) | 962 | 4,669 |
| Costs per perinatal life years gained | 37,914 | 118·4 |
| Costs per maternal life years gained | 18,374 | 244·4 |
| Costs per life years gained, perinatal and maternal combined | 56,288 | 79·7 |
| Costs per DALY averted, perinatal | 18,508 | 242·7 |
| Costs per DALY averted, maternal | 12,713 | 353·2 |
| Costs per DALY averted, perinatal and maternal combined | 31,221 | 143·8 |

SBBC = Safer Births Bundle of Care; DALY = Disease adjusted life years

**Table 3. Sensitivity analysis results.**

|  | Costs per life saved in USD | Costs per life year gained in USD | Costs per DALY averted in USD* | Costs-effectiveness according to WHO |
|---|---|---|---|---|
| No parameters altered | 4,469 | 80 | 144 | High |
| Lower limit of lives saved (225 newborns + 138 maternal lives) | 12,372 | 210 | 390 | High |
| Upper limit of lives saved (935 newborns + 626 maternal lives) | 2,877 | 49 | 89 | High |
| Double labour costs (= total costs of 6,899,905 USD) | 7172 | 122 | 221 | High |
| Treble labour costs (= total costs of 9,308,605 USD) | 9676 | 165 | 298 | High |
| Research costs omitted (= total costs of 3,664,822 USD) | 3809 | 65 | 117 | High |

All costs rounded to whole USD. *When discounting rate (r) = 0.03 per year and age weighting with K = 1 and β = 0.04;

DALY=Disease adjusted life years; WHO=World Health Organization.

## 4 Discussion

This retrospective evaluation of the costs and cost-effectiveness of SBBC Phase I showed that total costs incurred were 4,139,640 USD., while costs per life year gained were 80 USD and 144 USD per DALY averted, and the BCR was 36. Thus, the project has to be classified as highly cost-effective.

Our findings are in line with previous studies that found the cost-effectiveness of reproductive health interventions aimed at intrapartum care in low-income countries ranged from 200 to 400 USD per DALY, while a study from India reported costs of 205 USD per DALY averted and 5865 USD per life saved [20]. Furthermore, Sternberg et al. report that bundles of intra- and postpartum care can have a cost efficiency in the range of about 60–100 USD per DALY averted [4]. Packages or bundles that combine different interventions have proven more cost-effective than single interventions only targeting one specific task [6]. The BCR in these packages of reproductive health interventions, when scaled to a 90% coverage of the population, was estimated to be as high as 87 [21]. This number is considerably higher than the BCR of 36 reported in this study. However, as SBBC Phase I only covered 30 health facilities with approximately 300,000 deliveries and thus a small share of the population, the BCR might change with the scaling up of the programme.

From the total amount spent in SBBC Phase I, 21% was classified as start-up costs, 13% was allocated to research, and 66% was running costs. These results are similar to a cost analysis on an intervention in mother and child healthcare in India [22]. In addition, this study has similarities with the proportion of cost drivers, such as labour costs, which

amounted to 53% in our study as compared to 62% in the study by Prinja et al., and travel costs which represented 10% as compared to 15% of the latter [23].

Following the promising midterm results of the SBBC Phase I, the intervention was scaled up from 30 to 150 health facilities within the same regions in Phase II. Further economic evaluations will have to show whether cost-effectiveness is retained or might be even improved due to lower start-up and research costs and a higher coverage of the population, or if it will be reduced due to impaired sustainability of the outcomes over time of reduced effectiveness in sites with a lower burden of maternal and perinatal mortality and lower volumes of deliveries.

## 4.1 Study strengths and limitations

The SBBC Phase 1 was a multi-centre, prospective and observational study, with high-quality outcome data collected. In addition, it provided high statistical power due to a large sample size of almost 300,000 mother–newborn dyads observed [10]. Finally, the study settings were heterogeneous including health facilities in rural, semi-urban and urban settings, thus increasing the transferability of the study findings to other settings.

However, as discussed in more detail by Kamala and Ersdal et al., the study might suffer from incomplete or under-reported baseline data such as the number of fresh stillbirths, which could affect the evaluation of the effectiveness, and result in an underestimation of the number of lives saved [10,24,25]. The collection of cost data was performed retrospectively, based on the recordings of the accounting centre, and thus important information might not be captured, though the results from various sources were cross-checked to ensure validity. In addition, this study is merely a partial economic evaluation as it only deals with the programme's perspective, not capturing all costs arising to the health care provider like staff time and utilities, and leaving behind the societal part of economic evaluation. As we could not disaggregate the costs into costs allocated to newborn versus maternal care, the presented cost-effectiveness results for perinatal and mothers separately are of a rather theoretical value. As we lacked data on morbidity, DALYs were calculated based solely on years of life lost due to premature mortality (YLL) based on life expectancy in Tanzania, while we did not include years of healthy life lost due to disability (YLD). Instead, we assumed that there was no additional morbidity caused by saving the lives of newborns and mothers and no morbidity averted. This might have resulted in both an over- or underestimation in DALYs averted. Our findings of costs per DALY averted have thus to be treated with caution. However, our approach is in the same range as previous studies, where the number of DALYs averted in low-income countries was estimated as either half of the life years gained or 32 DALYs per life saved [5]. Also, a one-way SA might underestimate the overall uncertainties of parameters and ignore the correlation between parameters and hence not provide information about which parameters critically impact cost-effectiveness. Lastly, as the study lacks information on the costs of standard care, an incremental cost-effectiveness ratio could not be provided.

## 4.2 Policy implication

The SBBC Phase I appeared to be feasible in terms of financial perspectives and it is aligned with the national pro-grammes for reproductive and child health regarding training design, health commodities, technology, and the health information system as well as other health system building blocks [6]. In addition, our results show that Phase I was highly cost-effective as the SBBC combines different interventions, thus lowering the costs [10]. The bundle addresses not only the challenges of the national reproductive and child health programme of the Ministry of Health in the United Repub-lic of Tanzania but also the global efforts to achieve universal health coverage and sustainability goals 3.1 and 3.4 [26]. The bundle is user-friendly in the different levels of the healthcare system such as health centers, district hospitals and regional referral hospitals, and particularly in rural settings where most of the deliveries take place [22]. Rural settings are likely to have poorer coverage of quality health services and a higher share of home deliveries due to inadequate healthcare provision such as skilled staff or lack of the necessary infrastructure and equipment [10,27]. The design of the bundle that enables the local staff to perform continuous simulation-based training to retain their skills renders the

programme more self-sufficient and thus less dependent on continuous financial aid [25]. The training cascade with a strong anchorage at the local health facilities with low-cost and easy-to-maintain training and treatment equipment will facilitate the implementation of the bundle in other LMICs. As of today, it is implemented in Nigeria, but outcome data is not yet available.

### 4.3 Conclusion

The analysis of the SBBC Phase I intervention showed that it was highly cost-effective. In addition, it was feasible from a financial perspective and aligned with Tanzania's national programme for reproductive and child health. Currently, the bundle is being scaled up in Phase II, financed by the World Bank. Phase II implements the full bundle, including a training cascade, simulation-based training, the introduction of innovative tools, and a feedback loop based on local outcome data. However, as the scale-up also extends to health facilities with a lower burden of maternal and perinatal mortality and lower volumes of deliveries, efficiency may be altered. A nationwide scale up is planned, but awaits clarification of funding availability. Health-economic evaluations during these scale-ups must provide information on compliance with the original programme content, its clinical and financial sustainability, and the robustness of its cost-effectiveness.

## Author contributions

**Conceptualization:** Boniphace Marwa, Corinna Vossius, Hege Langli Ersdal.

**Data curation:** Boniphace Marwa, Corinna Vossius.

**Formal analysis:** Boniphace Marwa, Corinna Vossius, Jan Terje Kvaløy.

**Investigation:** Boniphace Marwa.

**Methodology:** Boniphace Marwa, Corinna Vossius.

**Supervision:** Corinna Vossius, Pascal Mdoe, Albino Kalolo, Domenica Morona.

**Validation:** Corinna Vossius, Frida N Ngalesoni, Albino Kalolo, Domenica Morona.

**Visualization:** Boniphace Marwa.

**Writing – original draft:** Boniphace Marwa, Corinna Vossius, Domenica Morona.

**Writing – review & editing:** Boniphace Marwa, Corinna Vossius, Benjamin Kamala, Pascal Mdoe, Frida N Ngalesoni, Albino Kalolo, Domenica Morona.

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
