## [Decision Letter · Decision Letter 0]

23 May 2025

Dear Dr. Marwa,

Thank you for submitting your manuscript to PLOS ONE. After careful consideration, we feel that it has merit but does not fully meet PLOS ONE’s publication criteria as it currently stands. Therefore, we invite you to submit a revised version of the manuscript that addresses the points raised during the review process.

We look forward to receiving your revised manuscript.

Kind regards,

Christopher Njeh

Academic Editor

PLOS ONE

2. Please upload a copy of the original ethics approval letter issued by the IRB for your study. If the original letter is not in English, please also provide an English translated version in the supporting file.

Also please report in the Methods section the dates when data were accessed for research purposes.

[The author received an unconditional grant from Laerdal Foundation for PhD studies.].

4. For studies involving third-party data, we encourage authors to share any data specific to their analyses that they can legally distribute. PLOS recognizes, however, that authors may be using third-party data they do not have the rights to share. When third-party data cannot be publicly shared, authors must provide all information necessary for interested researchers to apply to gain access to the data. (https://journals.plos.org/plosone/s/data-availability#loc-acceptable-data-access-restrictions)

6. Please amend either the abstract on the online submission form (via Edit Submission) or the abstract in the manuscript so that they are identical.

7. Please include a separate caption for each figure in your manuscript.

Additional Editor Comments:

Please address this

Limited Consideration of Morbidity: DALYs were estimated but disability weights were not applied, as morbidity data were not available. An argument of potential underestimation of averted DALYs would be more transparent.

Attribution of Outcomes: Even while the effectiveness is attributed to the SBBC intervention, clarification regarding potential confounders or co-occurring health interventions occurring over the period of study would add to causal inference.

Generalizability: Even if being used across varying levels of facilities and geographic areas, the manuscript could better clarify how findings would extend outside of the Tanzanian context, specifically to other LMICs that have varying capabilities of the health system.

Future Implications: The conclusion would be stronger if more focus were given to detailing anticipated Phase II and III expectations, especially financing mechanisms and health system integration.

This is an unambiguous, timely, and policy-relevant paper demonstrating SBBC Phase I to be overwhelmingly cost-effective according to WHO standards. It makes a significant contribution to the evidence base on maternal and neonatal health care interventions in LMICs. The paper is ready for publication after minor revisions, particularly to place the limitations and implications.

Reviewers' comments:

Reviewer's Responses to Questions

**Comments to the Author**

1. Is the manuscript technically sound, and do the data support the conclusions?

Reviewer #1: Yes

Reviewer #2: Yes

2. Has the statistical analysis been performed appropriately and rigorously?

Reviewer #1: No

Reviewer #2: Yes

3. Have the authors made all data underlying the findings in their manuscript fully available?

Reviewer #1: Yes

Reviewer #2: Yes

4. Is the manuscript presented in an intelligible fashion and written in standard English?

Reviewer #1: Yes

Reviewer #2: Yes

Reviewer #1: 99-101 ‘Cost data were based on secondary data collected retrospectively in March 2024. Data sources were the itemized budget of the project activities and the actual spending on the itemized activities as registered in the financial records’

– Would need proof that the itemized budget was not over-estimated in the first place. This is due to the varied pricing of goods/services in different countries e.g. the price of a pencil, for example, in the US differs greatly from its price in Tanzania. Also, was the intervention standard in all cases or were there some cases where less was done? This can affect the cost perceived as well. This is not clear.

119,120 ‘From the mothers, data were collected on maternal death, age, parity and whether she was referred from another hospital’

– What is the significance of the last variable? It is not mentioned anywhere else in the study.

169,170 ‘The total estimated number of lives saved during SBBC Phase I was 580 (95% CI 225, 935) perinatal lives 382 (95% CI 138, 626) maternal lives, and 962 combined lives saved’

– Please correct the punctuation in this sentence.

– There is no mention of the baseline that existed before the intervention. This raises the question whether the trend was worse before the intervention or not.

– Were other confounding factors taken into consideration such as non-partum related deaths such as stage 4 cancer or COVID-19 in the mother? These may have increased the number of maternal deaths despite the causes not being related to delivery.

269, 271 ‘In addition, this study is merely a partial economic evaluation as it only deals with the program’s perspective, not capturing all costs arising to the health care provider like staff time and utilities and leaving behind the societal part of economic evaluation.’

– Although mentioned in your study limitations, it would have been important, especially to inform policymakers, of the overall cost the intervention carries as the cost-benefits implied might be overshadowed by the overall economic cost.

Reviewer #2: Review of Manuscript: Cost-Effectiveness Analysis of the Safer Births Bundle of Care (SBBC) Phase I in Tanzania

Summary

This article reports a cost-effectiveness analysis of the Safer Births Bundle of Care (SBBC) Phase I, an intervention in maternal and neonatal health conducted in 30 Tanzanian health facilities from 2021 to 2023. The analysis uses strong outcome and expenditure data to estimate costs per life saved, life year gained, and DALY avoided, and to present a very favourable benefit–cost ratio (BCR) of 36.

Strengths

Policy Relevance: The research is extremely aligned with SDGs3.1–3.4, addressing an extremely high priority for health systems in low-resource settings.

Strong Methodology: Clear cost stratification (start-up, recurrent, research), use of documented cost-effectiveness thresholds, and the inclusion of exhaustive outcome measures like DALYs and life years gained strengthen the analysis.

High Sample Size: Analysis of nearly 300,000 mother–baby dyads provides a strong degree of statistical power and generalizability.

Transparency of Costing: Inclusive budget reporting and sensitivity analysis enhance replicability and credibility of policy.

Weaknesses and Recommendations

Limited Consideration of Morbidity: DALYs were estimated but disability weights were not applied, as morbidity data were not available. An argument of potential underestimation of averted DALYs would be more transparent.

Attribution of Outcomes: Even while the effectiveness is attributed to the SBBC intervention, clarification regarding potential confounders or co-occurring health interventions occurring over the period of study would add to causal inference.

Generalisability: Even if being used across varying levels of facilities and geographic areas, the manuscript could better clarify how findings would extend outside of the Tanzanian context, specifically to other LMICs that have varying capabilities of the health system.

Future Implications: The conclusion would be stronger if more focus were given to detailing anticipated Phase II and III expectations, especially financing mechanisms and health system integration.

This is an unambiguous, timely, and policy-relevant paper demonstrating SBBC Phase I to be overwhelmingly cost-effective according to WHO standards. It makes a significant contribution to the evidence base on maternal and neonatal health care interventions in LMICs. The paper is ready for publication after minor revisions, particularly to place the limitations and implications.

Recommendation: Minor revisions prior to acceptance.

.

Reviewer #1: No

Reviewer #2: **Yes:**Prof Abba MallumProf Abba MallumProf Abba MallumProf Abba Mallum

---

## [Author Response · Author response to Decision Letter 1]

23 Jun 2025

Academic Editor and reviewers comments and responses:

- PLOS ONE’s style requirements have been observed and adhered.

2. Please upload a copy of the original ethics approval letter issued by the IRB for your study. If the original letter is not in English, please also provide an English translated version in the supporting file.

- IRB for both National Institute for Medical Research (NIMR) and that of CUHAS have been uploaded.

Also, please report in the Methods section the dates when data were accessed for research purposes.

- The data were accessed between on March 2024, line ‘100’.

[The author received an unconditional grant from Laerdal Foundation for PhD studies.].

a)Please clarify the sources of funding (financial or material support) for your study. List the grants or organizations that supported your study, including funding received from your institution.

-The author received an unconditional research grant from Laerdal Foundation. There was no any material support from the funder.

- The funder had no role in study design, data collection and analysis, decision to publish, or preparation of the manuscript.

-Corrina Vossius received a salary from the SBBC program, that was funded by the Global Facility Financing (GFF). Other authors received no specific funding for this work.

- No any authors received a salary from any of my funders.

-Other authors received no specific grant/salaries for this work.

-Corrina Vossius received a salary from the SBBC program, that was funded by the Global Facility Financing (GFF). Other authors received no specific funding for this work.

4. For studies involving third-party data, we encourage authors to share any data specific to their analyses that they can legally distribute. PLOS recognizes, however, that authors may be using third-party data they do not have the rights to share. When third-party data cannot be publicly shared, authors must provide all information necessary for interested researchers to apply to gain access to the data. (https://journals.plos.org/plosone/s/data-availability#loc-acceptable-data-access-restrictions)

1)A description of the data set and the third-party source

- The cost data were collected from Haydom Lutheran Hospital accounting office as described in Manuscript between line ‘100 -104’. Outcome data for maternal and newborn born are described in the article of Benjamin et al., Outcomes of a Program to Reduce Birth-Related Mortality in Tanzania, DOI: 10.1056/NEJMoa2406295.

-The data can be requested from Haydom Lutheran Hospital, P.O.BOX 9000 Haydom, Manyara -Tanzania.

Tel. +255(0)27 253 3194/5, Fax +255(0)27 253 3734, E-mail post@haydom.co.tz

-There were no special privileges the authors received.

- The contact information for gaining access to the data is stated under 4.5 Data sharing statement.

-The pre-existing an ORCID iD have been authenticated in Editorial Manager.

6.Please amend either the abstract on the online submission form (via Edit Submission) or the abstract in the manuscript so that they are identical.

-Abstract on the online submission and that in the manuscript have been amended and they are identical.

7. Please include a separate caption for each figure in your manuscript.

-Separate captions for each figure have been included in the submitted manuscript.

Response to Reviewers

These requirements are now been fulfilled.

Comments from the Editor and Reviewer #2:

Thank you very much for your positive evaluation.

1. Limited Consideration of Morbidity: DALYs were estimated but disability weights were not applied, as morbidity data were not available. An argument of potential underestimation of averted DALYs would be more transparent.

Answer. We have now rephrased the discussion of the limitations in calculating DALYs averted. We reckon that morbidity might have been estimated both too low, (as asphyxia might cause brain damage that we have not accounted for), and too high (as the intervention might have prevented cases of asphyxia). Therefore, we might have both over- and underestimated the number of DALYs averted by not taking YLDs into account. (Discussion, lines ‘280-289’)

2. Attribution of Outcomes: Even while the effectiveness is attributed to the SBBC intervention, clarification regarding potential confounders or co-occurring health interventions occurring over the period of study would add to causal inference.

Answer: We have now clarified in the description of the SBBC intervention, that the SBBC implementation was the only relevant change to the health care system and treatment recommendation at this time (Introduction, page 2,’85-86’)

3. Generalizability: Even if being used across varying levels of facilities and geographic areas, the manuscript could better clarify how findings would extend outside of the Tanzanian context, specifically to other LMICs that have varying capabilities of the health system.

Answer: We have now added a paragraph about generalizability under Policy implication, where we state as well that the bundle is presently implemented in Nigeria (Discussion, Policy implication, page 14,’307-313’).

4. Future Implications: The conclusion would be stronger if more focus were given to detailing anticipated Phase II and III expectations, especially financing mechanisms and health system integration.

Answer: We have now added the suggested details about the present Phase II and planned Phase III in the Conclusion. (Conclusion, ‘318-326’)

Comments from Reviewer #1:

1. 99-101 ‘Cost data were based on secondary data collected retrospectively in March 2024. Data sources were the itemized budget of the project activities and the actual spending on the itemized activities as registered in the financial records’

– Would need proof that the itemized budget was not over-estimated in the first place. This is due to the varied pricing of goods/services in different countries e.g. the price of a pencil, for example, in the US differs greatly from its price in Tanzania. Also, was the intervention standard in all cases or were there some cases where less was done? This can affect the cost perceived as well. This is not clear.

Answer: Thank you for pointing out this important topic. The budget was compiled by the Tanzanian primary investigator and his team. It was originally expressed in Tz shilling, and only for the purpose of this paper converted to USD. To emphasize this, we stated that “All costs were estimated in Tanzanian Shillings (TZS) and converted to 2020 USD using a mean exchange rate of TZS 2,305/USD provided by the Central Bank of Tanzania” in the section Methods – Cost data collection (page 3, ‘111-113’) Insight into the budget and the actual spendings can be requested at the Haydom Lutheran Hospital, as stated in the Data sharing statement.

119,120 ‘From the mothers, data were collected on maternal death, age, parity and whether she was referred from another hospital’

– What is the significance of the last variable? It is not mentioned anywhere else in the study.

Answer: The variable about whether or not the mother was referred from another hospital was used in the statistical analysis of the death incidences, as data was adjusted for clinically relevant demographic and clinical factors. This is described in detail in Kamala and Ersdal at al. Outcomes of a Program to Reduce Birth-Related Mortality in Tanzania. New England Journal of Medicine. 2025.

169,170 ‘The total estimated number of lives saved during SBBC Phase I was 580 (95% CI 225, 935) perinatal lives 382 (95% CI 138, 626) maternal lives, and 962 combined lives saved’

– Please correct the punctuation in this sentence.

– There is no mention of the baseline that existed before the intervention. This raises the question whether the trend was worse before the intervention or not.

– Were other confounding factors taken into consideration such as non-partum related deaths such as stage 4 cancer or COVID-19 in the mother? These may have increased the number of maternal deaths despite the causes not being related to delivery.

Answer: Thank you for pointing to that important question. We have not considered non-partum related deaths. However, as this study observes mortality during birth, this will probably not be affected by severe diseases like cancer or other non-communicable diseases. As the study period lasted from 2021 to 2023, concurrent with the COVID pandemic, this is unlikely to impact the study outcome. As to best of our knowledge, there were no other epidemics in the observed regions during that time.

269, 271 ‘In addition, this study is merely a partial economic evaluation as it only deals with the program’s perspective, not capturing all costs arising to the health care provider like staff time and utilities and leaving behind the societal part of economic evaluation.’

– Although mentioned in your study limitations, it would have been important, especially to inform policymakers, of the overall cost the intervention carries as the cost-benefits implied might be overshadowed by the overall economic cost.

Answer: We absolutely agree with the Reviewer. The study group has just received a grant to explore the full scope of health economic consequences of the program and its spill over effects. Unfortunately, the present paper had to rely on the available information.

---

## [Decision Letter · Decision Letter 1]

28 Nov 2025

Dear Dr. Marwa,

We look forward to receiving your revised manuscript.

Kind regards,

Denny John

Academic Editor

PLOS ONE

Journal Requirements:

Reviewers' comments:

Reviewer's Responses to Questions

**Comments to the Author**

Reviewer #1: All comments have been addressed

Reviewer #3: (No Response)

2. Is the manuscript technically sound, and do the data support the conclusions?

Reviewer #1: Yes

Reviewer #3: Yes

3. Has the statistical analysis been performed appropriately and rigorously?

Reviewer #1: Yes

Reviewer #3: Yes

4. Have the authors made all data underlying the findings in their manuscript fully available?

Reviewer #1: Yes

Reviewer #3: (No Response)

5. Is the manuscript presented in an intelligible fashion and written in standard English?

Reviewer #1: Yes

Reviewer #3: Yes

Reviewer #1: All the reviewer's comments were addressed in an adequate manner. No further comments to the Author.

Reviewer #3: The article “Costs and Cost-Effectiveness of the Safer Births Bundle of Care Intervention in Tanzania Health Facilities” presents an important analysis of a phase 1 implementation of the Safer Births Bundle of Care in 30 health facilities in Tanzania from 2021 to 2023. This article presents data that was collected retrospectively. While the article mentions some information about the intervention in lines 71-78 it is difficult to understand what exactly is involved in this bundle of care. While this is an economic analysis it would be helpful to have more detail about what was involved in the intervention to give this more context.

It would also be of interest to understand how many people were involved in the training and implementation of this across these centers. Interestingly this study was done in both rural and non-rural areas with different levels of deliveries. It would be interesting if they could present some of the information on cost and cost-effectiveness in these different types of settings.

They discuss travel cost being limited due to the pandemic restrictions. It would be helpful to understand a bit more about whether the intervention facilitated travel within the region and also affected women to get referred from primary to secondary to tertiary centers if required. It would be helpful to understand the regional context a bit. Could emergency C-sections be performed in the primary center and did they have newborn resuscitation facilities in these primary centers? It would be interesting to understand how this intervention worked and the costs and cost-effectiveness in the primary vs. secondary vs. tertiary setting.

This is a commendable study and should be accepted with minor corrections.

.

Reviewer #1: No

Reviewer #3: No

---

## [Author Response · Author response to Decision Letter 2]

10 Dec 2025

Dear Editor and Reviewers,

Kindly receive the responses to the comments of Reviewer #3 to the manuscript numbered ‘PONE-D-25-18416’ entitled "Costs and Cost-Effectiveness of the Safer Births Bundle of Care Intervention in Tanzania Health Facilities."

First and foremost, we thank the editor for taking over the publication process and handling the manuscript in a timely manner.

Secondly, we want to thank the reviewers for revising the manuscript and provide their thoughtful comments and constructive critique that would help to improve the quality and clarity of our manuscript.

We are very happy that Reviewer #1 has no further comments.

Comments from Reviewer #3:

Thank you very much for your positive evaluation.

1. While the article mentions some information about the intervention in lines 71-78 it is difficult to understand what exactly is involved in this bundle of care. While this is an economic analysis it would be helpful to have more detail about what was involved in the intervention to give this more context. It would also be of interest to understand how many people were involved in the training and implementation of this across these centers.

Answer: We have now given more detail about the main clinical targets of the intervention and the organization of the initial training as cascade training (Introduction, The Safer Births Bundle of Care, page 2, line 79 – 85.)

2. Interestingly this study was done in both rural and non-rural areas with different levels of deliveries. It would be interesting if they could present some of the information on cost and cost-effectiveness in these different types of settings.

Answer: This is indeed a very relevant question and similar to question 5, where you enquire about a sub-analysis of the different levels of healthcare services. Unfortunately, both topics are beyond the scope of this paper. However, we are currently planning to perform these sub-analyses.

3. They discuss travel cost being limited due to the pandemic restrictions. It would be helpful to understand a bit more about whether the intervention facilitated travel within the region and also affected women to get referred from primary to secondary to tertiary centers if required. It would be helpful to understand the regional context a bit.

Answer: We have now clarified that the travel restrictions due to the Corona-pandemic concerned the national and international faculty that was conducting and supervising the training. The travels of the mothers to the hospitals or during referrals was not part of the SBBC budget, and we have no data about it. However, regional travels in Tanzania were not restricted during the pandemic. (Results, Costs of the SBBC intervention, page 7, line 194)

4. Could emergency C-sections be performed in the primary center and did they have newborn resuscitation facilities in these primary centers?

Answer: We have now added the information that all of the health facilities provide Comprehensive Emergency Obstetric and Basic Newborn Care services and have separate labour units and operating theatres. Before the start of the intervention, we assessed the readiness of the health facilities and refurbished them, where necessary. (Introduction, The Safer Births Bundle of Care, page 2, line 90-92)

5. It would be interesting to understand how this intervention worked and the costs and cost-effectiveness in the primary vs. secondary vs. tertiary setting.

Answer: Please see our comment to question 2.

Regards,

Boniphace Richard Marwa,

Corresponding Author

---

## [Editor Report · Decision Letter 2]

9 Mar 2026

Costs and Cost-Effectiveness of the Safer Births Bundle of Care Intervention in Tanzania Health Facilities

PONE-D-25-18416R2

Dear Dr. Marwa,

We’re pleased to inform you that your manuscript has been judged scientifically suitable for publication and will be formally accepted for publication once it meets all outstanding technical requirements.

Kind regards,

Tanya Doherty, PhD

Academic Editor

PLOS One

Additional Editor Comments (optional):

I suggest adding to the limitations section that you have not presented costs for rural versus urban facilities or different levlels of care (primary vs tertiary) with different delivery volumes since this would impact cost and cost efficiency.
---

## [Editor Report · Acceptance letter]

PONE-D-25-18416R2

PLOS One

Dear Dr. Marwa,

I'm pleased to inform you that your manuscript has been deemed suitable for publication in PLOS One. Congratulations! Your manuscript is now being handed over to our production team.

Kind regards,

on behalf of

Professor Tanya Doherty

Academic Editor

PLOS One